# Gut-Derived Uremic Toxins in CKD: An Improved Approach for the Evaluation of Serum Indoxyl Sulfate in Clinical Practice

**DOI:** 10.3390/ijms24065142

**Published:** 2023-03-07

**Authors:** Gianvito Caggiano, Loredana Amodio, Alessandra Stasi, Nicola Antonio Colabufo, Santina Colangiulo, Francesco Pesce, Loreto Gesualdo

**Affiliations:** 1Nephrology, Dialysis and Transplantation Unit, Department of Precision and Regenerative Medicine and Ionian Area (DiMePRe-J), University of Bari “Aldo Moro”, 70122 Bari, Italy; 2Biofordrug S.R.L., University of Bari Spin-Off, 70019 Triggiano, Italy; 3Department of Pharmacy-Drug Sciences, University of Bari “Aldo Moro”, 70122 Bari, Italy

**Keywords:** CKD, hemodialysis, uremic toxins, indoxyl sulfate, LC-MS/MS

## Abstract

In the past years, indoxyl sulfate has been strongly implicated in kidney disease progression and contributed to cardiovascular morbidity. Moreover, as a result of its elevated albumin affinity rate, indoxyl sulfate is not adequately cleared by extracorporeal therapies. Within this scenario, although LC-MS/MS represents the conventional approach for IS quantification, it requires dedicated equipment and expert skills and does not allow real-time analysis. In this pilot study, we implemented a fast and simple technology designed to determine serum indoxyl sulfate levels that can be integrated into clinical practice. Indoxyl sulfate was detected at the time of enrollment by Tandem MS from 25 HD patients and 20 healthy volunteers. Next, we used a derivatization reaction to transform the serum indoxyl sulfate into Indigo blue. Thanks to the spectral shift to blue, its quantity was measured by the colorimetric assay at a wavelength of 420–450 nm. The spectrophotometric analysis was able to discriminate the levels of IS between healthy subjects and HD patients corresponding to the LC-MS/MS. In addition, we found a strong linear relationship between indoxyl sulfate levels and Indigo levels between the two methods (Tandem MS and spectrophotometry). This innovative method in the assessment of gut-derived indoxyl sulfate could represent a valid tool for clinicians to monitor CKD progression and dialysis efficacy.

## 1. Introduction

As a result of chronic kidney disease (CKD) progression, the gradual loss of the glomerular filtration rate represents the driving force for the accumulation of toxic solutes involved in uremic syndrome. The hefty retention of these compounds in the body contributes to detrimental consequences and is strongly associated with poor outcomes in CKD patients [1,2]. In the past years, a considerable number of harmful molecules have been identified and grouped in line with their size, structure, chemical properties and ability to bind plasma proteins [3]. According to this classification, the first cluster of molecules is represented by small toxins (e.g., creatinine and urea), which are successfully cleared by traditional dialysis methods. The second group comprises middle-weight solutes such as large peptides, proteins, and lipoproteins (e.g., b2-microglobulin, parathyroid hormone, cytokines, etc.). Finally, due to their high affinity with blood proteins, a further class is represented by protein-bound uremic toxins (PBUTs) [4]. Of note, PBUTs include gut-derived compounds that originate from the proteolytic metabolism of intestinal microbiota. In detail, as a result of the decrease in renal filtration rate, urea accumulates in the intestinal lumen [5]. This uremic milieu supports the overgrowth of harmful microbial communities such as tyrosinase- and tryptophanase-positive bacteria. Consequently, the large retention of indoles, cresols, and phenols drives intestinal permeability, resulting in the translocation of such toxins into the blood [6,7,8]. A growing amount of data demonstrates that the elevated plasma concentration of PBUTs causes multi-organ damage.

Notably, several PBUTs, including indoxyl sulfate (IS) and p-Cresyl Sulfate (pCS), are of particular interest for their detrimental effects associated with the exacerbation of renal damage and CKD-associated complications such as renal inflammation and fibrosis, cardiovascular morbidity, atherosclerosis thrombosis, and mortality. For instance, several in vivo and in vitro studies indicated that IS and pCS were implicated in vascular damage by inducing oxidative stress, senescence, and apoptosis in endothelial cells [9,10,11,12,13]. Moreover, at the renal level, indoles and cresols play a pivotal role in triggering intrarenal inflammation and tubulo-interstitial fibrosis by modulating several intracellular signals, including p53, AhR, Nf-kB, and klotho [14,15,16,17]. On the other hand, due to their strong pro-oxidant effect, the protein-bound solutes were shown to drive metabolic dysfunction by inducing insulin resistance, sarcopenia, and adipocyte injury [18,19,20].

IS levels are associated with poor prognosis, especially during the late stages of kidney disease [21,22,23]. In particular, the plasma levels of IS progressively increase from stage I to stage V, reaching a maximum in end-stage kidney disease (ESKD), indicating that the clearance of IS gradually decreases with the advanced decline of kidney function [23,24]. In addition, hemodialysis (HD) is unable to purify the blood from PBUTs since the plasma–protein bond hinders their clearance [24,25]. The introduction of high-flux membranes (HF) and the use of hemodiafiltration (HDF) improved the middle molecules’ removal by relying on larger pore sizes and increasing the convection mechanism. However, potential albumin and nutrient loss might occur and should be taken into account with these methods; in addition, they have limited effect on removing PBUTs [4,26]. As a consequence of this accumulation, HD patients exhibit elevated serum concentrations of indoxyl sulfate, resulting in detrimental conditions, including cardiovascular damage and poor survival rates [27]. In this scenario, the quantification of IS could aid clinicians in monitoring both the CKD progression and the efficiency of the dialysis treatment. Moreover, the early quantification of IS could represent a good strategy for predicting CKD-related complications such as cardiovascular events. 

Conventionally, liquid chromatography–tandem mass spectrometry (LC-MS/MS) represents the main diagnostic system in the measurement of IS thanks to its elevated performance [28,29,30,31]. Nevertheless, it requires dedicated skills and costly equipment, and results are obtained after a long period. In recent years, several working groups have improved the LC-MS/MS method for the quantification of blood IS. However, most of these methods still require laborious sample preparation and various steps to be carried out [32,33]. In this study, we aimed to develop a novel approach for rapidly estimating the blood’s indoxyl sulfate levels in CKD patients that could be implemented in clinical practice. Importantly, compared to the current systems, Theremino represents the new frontier in the context of spectrophotometry-based devices that can be used in clinical practice. In detail, Theremino is a smart spectrophotometer thanks to its compatibility with tablets, PCs, and smartphones since it is equipped with a USB connection. Therefore, the data can be quickly transmitted in a dataset or to a control room. Moreover, thanks to its small size (40 × 10 cm) and light weight (500 g), it is easily transportable to where the test will take place (i.e., outpatient or inpatient rooms) and can acquire data within a selected UV-Vis spectrum. In contrast, currently used spectrophotometric instruments are expensive and cannot be used in a clinical setting since some are rather unwieldy and heavy and require specific skills or laborious protocols. On this basis, considering that Indigo formation occurs in a few minutes and its spectrum is promptly recorded, it is possible to consider Theremino as a first-line tool for detecting the levels of indoxyl sulfate in the CKD population.

To the best of our knowledge, no study has implemented the detection of this uremic toxin without using mass spectrometry as of yet. In addition, there are no published methods for IS quantification using a spectrophotometric assay for clinical settings.

## 2. Results

### 2.1. Indigo Derivatization from IS

The derivatization of IS by FeCl_3_ generates the Indigo-blue chromophore that is able to absorb light at λ = 450 nm. The summary of the derivatization reaction is shown in Figure 1. Briefly, we added 300 µL of Acetonitrile to the serum. After centrifugation, 100 µL of sulfuric acid solution (1:1) and 500 µL of FeCl_3_ 0.001 M (in HCl 1N) were added to the supernatant. The mixture was held for 15 min at 70 °C. Finally, Indigo was measured by both LC-MS/MS and Theremino.

### 2.2. Calibration and Linearity

Linear correlation coefficients (R) were determined based on the correlation between absorbance spectral values obtained with known concentrations of Indigo. The best wavelengths for estimating the concentration of Indigo were found by Lambert–Beer’s law: A(λ)= Ɛ M l, (A(λ) = UV absorbance value at certain wavelength; Ɛ = molar absorptivity coefficient for Indigo; M = molarity (mol/L)). 

A calibration curve was constructed by using six points for different concentrations of analytes (7.5, 15, 30, 50, 75, and 95 µM) (Figure 2). The calibration equation was A = 0.0039C + 0.017 (A = analyte absorption; C = relative concentration of the analyte). We confirmed the formation of Indigo with the derivatization reaction of indoxyl sulfate by LC-MS/MS analysis.

#### 2.2.1. Experimental Method of LC-MS/MS

##### Linearity

The correlation coefficient of R^2^ = 0.9986 (Figure 2) proved linearity over the concentration range. The equation of the calibration curve was y = 67.07x + 2.25 for IS and y = 973.04x + 539.94 for pCS, where y represents the peak area ratio of analytes to DHTC, and x represents the relative concentration of the analytes. 

##### Accuracy

Accuracy was evaluated at two different concentration levels (7.5 and 50 mg/mL) and with 9-fold injection. The accuracy was 97.6 ± 0.2% for 7.5 mg/mL and 99.2 ± 0.1% for 50 mg/mL of the analytical solution.

##### Precision

The limit of detection and the limit of quantification were calculated by equations LOD = 3.3 SD/s and LOQ = 10 SD/s, respectively. In the equation, s represents the slope of the calibration curve, and SD represents the standard deviation of the peak area. The acceptance criterion was LOD = 0.067 and LOQ = 0.204 for IS and LOD = 0.072 and LOQ = 0.218 for pCS. The intraday precision of the assay method was evaluated by carrying out nine independent assays of an analytic solution prepared at two concentration levels (7.5 and 50 mg/mL). For interday precision at each concentration level, a single injection of the solution was assayed daily for three consecutive days. The % relative standard derivations of intra- and interday assays were 0.5% for 7.5 mg/mL and 0.8% for 50 mg/mL, as well as 9.5% for 7.7 mg/mL and 8.7% for 50 mg/mL of the analyte. 

### 2.3. Method Comparison

Serum from both HD and healthy individuals was collected in order to measure IS and Indigo levels by both colorimetric assay (using the Theremino Spectrophotometer) and LC-MS/MS. IS levels measured by tandem mass spectrometry are shown in Figure 3A. As expected, the amount of serum IS in HD patients increased when compared to healthy controls. Next, from each serum sample, we performed the derivatization of the IS into Indigo by FeCl_3_ (Figure 1). At the end of each reaction, the total amount of IS-derived Indigo was measured by both a Theremino spectrophotometer and LC-MS/MS (Figure 3B,C). Notably, the levels of Indigo measured by Theremino were higher in the HD group compared with heathy people. Of note, these data indicate that the total amount of IS in the serum samples of HD patients completely reacted in producing Indigo blue. Additionally, the same result was obtained when Indigo from the same sample was quantified by standard Tandem MS. Importantly, the levels of both Indigo and IS in each group were highly comparable when measured by the Theremino spectrophotometer and LC-MS/MS, respectively (Figure 3D). Therefore, altogether, these results demonstrate that the new method based on Theremino was able to discriminate the levels of IS (using Indigo) between healthy subjects and HD patients corresponding to the LC-MS/MS. In order to evaluate differences between the free and protein-bound fractions, the amount of free indoxyl sulfate was measured by LC-MS/MS and Theremino. Interestingly, the quantification of IS-bound fractions observed with LC-MS/MS (bound form, >97 ± 1.9%) was comparable to the quantification of Indigo with Theremino (<94 ± 1.3%). Consequently, the unbound fractions of IS and Indigo were both less than 10%.

To investigate the difference between the two methods, we related the levels of IS measured by LC-MS/MS to the levels of Indigo quantified by the Theremino spectrophotometer (Figure 4A) using the serum samples of HD participants. A linear correlation was represented by the regression line y = 0.9x + 9.86 with a relationship (R^2^) of 0.951 (*p* < 0.001). Furthermore, there was a significant association between the concentration of Indigo detected by each method (Figure 4B) and the levels of IS and Indigo detected by LC-MS/MS alone (Figure 4C) (*p* < 0.001). Finally, a congruent agreement was found between the quantification of indoxyl sulfate by the standard approach and the levels of Indigo detected by Theremino (Figure 4D).

## 3. Discussion

As a result of kidney failure, CKD patients developed a strong retention of different uremic compounds, which are implicated in the advancement of kidney injury. In this scenario, HD patients are characterized by the highest levels of these toxins, including IS and pCS, which are closely related to poor prognosis [34,35]. Moreover, a large number of clinical reports also suggest that the level of serum IS is able to predict the progression of CKD since its incremental accumulation is directly correlated to the gradual failure of renal function [36,37,38]. Moreover, animal and human studies observed that the elevated concentration of IS is tightly associated with all-cause mortality [39,40]. Finally, the total amount of gut-derived indoxyl sulfate was also able to predict the prevalence of CKD-related complications, including endothelial injury, cardiac failure, metabolic dysfunction, and bone disease [41,42,43]. Altogether, these data highlight the importance of IS as a novel potential biomarker in the context of CKD since the current strategies seem to be inadequate in predicting the CKD-associated comorbidities promoted by protein-bound uremic toxins. Therefore, the rapid measurement of IS concentrations through the CKD stages could represent a reliable index for clinicians in predicting adverse outcomes and also assessing adherence to nutritional therapy [1,40,44,45]. Additionally, the assessment of IS amounts in HD populations could be employed to better evaluate the efficiency of dialysis therapies.

Based on this statement, the evaluation of IS levels in CKD and ESKD patients can be employed as a measurement-of-risk rate with respect to kidney injury progression, CVD, and mortality [46,47]. Interestingly, it has been shown that during urethral infections, several sulfatase- and phosphatase-producing bacteria can cause urine color change to blue-purple. Interestingly, this biological process involves the enzymatic transformation of indoxyl sulfate into Indigo, which is promoted by microbic phosphatases or sulfatases [48]. The foundation of our work is the transformation of IS into Indigo by a chemical reaction and its spectrophotometric measurement.

Chiefly, tandem mass spectrometry represents a widely used diagnostic approach used for the Indole measurement thanks to its great performance in sensitivity [49,50,51]. This method allows for higher sensitivity and accuracy; however, it requires dedicated equipment and skills for analyses, and results are obtained after a long period. Hence, to counteract these issues, we developed a real-time approach based on a spectrophotometric assay that is exploitable in clinical settings.

We implemented chemical derivatizations in order to transform the total amount of IS into Indigo blue in serum samples from HD patients and healthy donors. This method allowed a rapid quantification of IS-derived Indigo at λ = 420–450 nm by using the mini-spectrophotometer Theremino. Moreover, to address the method’s feasibility, we compared the levels of Indigo measured with Theremino to the levels of IS and Indigo measured with LC-MS/MS. Both the specificity and the repeatability of Indigo levels monitored by Theremino were strongly correlated to the standard method. Since numerous studies show that the serum IS levels of healthy subjects are <10 μM and >20 μM for HD patients, we considered that the linearity together with the LLQ was satisfactory for sample measurements. Furthermore, our method was not influenced by external agents or other uremic toxins such as pCS, demonstrating its high specificity for IS. In our analysis, the spectrophotometric method showed that Indigo’s intensity showed a 3-fold increase in the patient group compared to the baseline amount. Remarkably, this result was closely comparable to the LC-MS/MS analysis of IS and IS-derived Indigo before and after derivatization, respectively. According to these data, the new method based on the Theremino spectrophotometer is characterized by high sensitivity in discriminating the levels of IS (by Indigo) between healthy subjects and HD patients corresponding to the standard method. Moreover, we found that the serum amount of the total and free fraction of indoxyl sulfate was highly comparable between LC-MS/MS and Theremino. This result was in line with previous data showing a high albumin-binding ratio of 90% [52].

Additionally, we observed a significant direct association between Indigo measured by the Theremino spectrophotometer and IS measured by LC-MS/MS in serum samples of HD patients (R^2^ = 0.9512, *p* < 0.001). Finally, the correlation between the two methods using the Bland–Altman Plot highlighted a profound agreement between the techniques. Furthermore, the reliability of this novel test was justified by high reproducibility and good accuracy and was not influenced by the matrix effect. We observed that the sensitivity of this assay was not affected by the quality of blood sample (e.g., cloudy serum, blood clots, or freeze/thaw cycles).

This study has some limitations. In particular, our analysis was carried out on a relatively small cohort of subjects (25 HD patients and 20 healthy individuals), and given the sample’s size, this could be considered a pilot study. In addition, this method is unable to simultaneously measure other biological markers, such as creatinine, albumin, or other toxins. It should also be noted that this method is unable to simultaneously measure further biological markers or toxins implicated in kidney pathophysiology. The colorimetric evaluation of IS-derived Indigo is not able to discriminate between the free and bound fraction of the analyte. Future studies are needed to split the free fraction of PBUT from the protein-bound fraction by using a differential separation of proteins in serum samples. 

It can be argued that although tandem MS is a high-ranking technique in the measurement of several toxins as well as the markers of CKD, it requires dedicated equipment and expert skills and does not allow real-time analysis. Our method allows size reductions and is cost-cutting. Moreover, this approach is easier and can be performed without the need of expert laboratory professionals. Importantly, the Theremino spectrophotometer can provide data in situ to promptly detect the level of IS in CKD and HD patients. Based on these data, clinicians will be able to evaluate the effects of treatments aimed at reducing uremic toxin levels, such as a low-protein diet, microbiota replacement, and oral adsorbent [17,53,54].

## 4. Materials and Methods

A multiresidue analysis by tandem MS was carried out in the estimation of the concentration of IS in the blood of both HD and the control group. Accordingly, the serum IS was derivatized by a rapid reaction in a stable chromophore called Indigo blue. The results of the chemical conversion of IS into Indigo were monitored by LC-MS/MS, demonstrating that the total amount of serum IS was completely converted into Indigo after 15 min. Ultimately, the total concentration of IS-derived Indigo was evaluated by using a smart UV-Vis spectrometer. Furthermore, we developed a smart device named “Theremino Spectrophotometer”, which can convert the absorbance level of Indigo (defined by the range of Indigo quantification from 10 mM to 400 mM in UV-VIS) in a concentration value of IS that can be digitally acquired and managed.

### 4.1. Patients

We enrolled a total of 20 healthy volunteers and 25 HD patients from the Nephrology Unit of the “Policlinico di Bari’’, Bari, Italy. 

The study has been carried out in accordance with the Helsinki Declaration and approved by the Ethical Committee of the “Azienda Ospedaliero Universitaria Consorziale Policlinico”, Bari, Italy (approval number: 6137). All participants provided written informed consent. We recruited 25 male patients with CKD stage V on standard hemodialysis, and they were all Caucasian and aged 18 years and older (median age: 56 years). Standard hemodialysis consisted of four-hour sessions for a total of three times per week (dialysis vintage ≥ 3 months). Exclusion criteria were as follows: acute kidney injury (AKI), history of heart failure, and previous peritoneal dialysis or renal transplantation. Serum was collected from total blood samples before the hemodialysis session and centrifugated at 3000× *g* for 15 min. 

### 4.2. Chemicals and Reagents

For Tandem MS, internal standards including indoxyl sulfate potassium salt and ammonium acetate (99% purity) were supplied by Sigma-Aldrich (product of Germany). Methanol (CH_3_OH) and acetonitrile (MeCN; CH_3_CN) were purchased from Merck (Darmstadt, Germany). For the synthesis of Indigo, we used iron (III) chloride anhydrous salt with purity > 97% (Fisher Chemicals-Waltham, MA, USA; Lot. 1915377); hydrochloric acid solution (1N) (Honeywell, Fluka, Lot. K0900); ethyl acetate (Honeywell, Fluka-Seelze, Germany; Lot. I3180); and sulfuric acid, 97% *v*/*v* (Sigma Aldrich-St. Louis, MO, USA; Lot. JO070).

### 4.3. Sample Pretreatment

The total amount of indoxyl sulfate was obtained by the centrifugation of MeCN with the serum (1:1, *v*/*v*) at 5000× *g* for a total of 5 min. Next, the supernatant was blended with 10mM of ammonium acetate buffer (1:1, *v*/*v*) and DHTC (2 μg/mL). In order to quantify the amount of the unbound form of indoxyl sulfate, the serum was ultrafiltered by centrifugation at 10.000× *g* for 30 min using a 3K MWCO filter to separate proteins from the whole serum. After deproteinization, the ultrafiltered serum was treated with 10mM of ammonium acetate buffer (1:1) and DHTC (2 μg/mL). Finally, the total and the free fraction of indoxyl sulfate were quantified by both LC-MSMS and Theremino.

### 4.4. LC-MS/MS Conditions of Uremic Toxins and Indigo

Uremic toxins and Indigo were analyzed using LCMS 8040 Triple Quadrupole LC/MS/MS (Shimadzu), as previously described [55].

Electrospray (ESI, negative mode at 4000 V) was performed for each analysis in LC-MS/MS. The mobile phase was composed of MeCN mixed with 2 mM of ammonium acetate. Nitrogen was employed for the nebulizer and desolvation gas and finally utilized as collision gas for molecule dissociation [28]. Notably, for Indigo determination, 10% MeCN was used for equilibrating the column (Luna Omega, size 150 × 4.6 mm, Phenomenex). Next, MeCN was raised up to 50% and finally scaled at 10% of MeCN [56]. The fragmentation voltages and collision energies of analytes for the first and second methods were optimized, as reported in Table 1. The chromatograms and MRM (multiple reaction monitoring) spectrum of IS, pCS, DHTC, and Indigo are shown in Figure 5.

### 4.5. Spectrometer Apparatus

The Theremino Spectrometer provides results based on the measurement of transmitted light by Indigo at λ = 410–570 nm. The Theremino System software calculates the light intensity derived from each pixel, resulting in the measurement of the amount of light emission. For Indigo, we measured the transmitted light in the range of λ = 420–450 nm. 

### 4.6. Stability

The sample’s freezing stability was determined by evaluating the concentration of analytes in each sample at +4° after 24 h and at −80° after 7 days and 30 days.

IS quantification with Theremino is based on spectrophotometry, thus relying on the capacity of IS to originate Indigo. Based on our analyses, we observed that the sensitivity of the Theremino was not dependent on blood rheology (e.g., cloudy serum or blood clots) as there was no interference with IS concentrations or the derivatization reaction. Moreover, we observed that sample quality did not influence IS reactions, such as Indigo generation or its absorbance at wavelengths of 420–450 nm, and evaluated the ability of Theremino to perform quantifications under different conditions, such as freeze/thaw cycles.

### 4.7. Statistical Analysis

GraphPad Prism 8.0 (GraphPad Software, La Jolla, CA, USA) and IBM SPSS Statistics v.25 (Apple MacOS) (IBM, Armonk, NY, USA) were used to perform correlations, regression analysis, and to obtain a Bland–Altman plot for method comparison. The statistical significance of each analysis was considered at a *p*-value < 0.05.

## 5. Conclusions

IS represents a hallmark in CKD progression and CVD-related risks in HD patients. Our innovative approach can provide rapid information regarding the serum concentration of IS in CKD and HD populations. The routine measurement of this toxin using our novel tool in clinical practice can be useful to better manage these patients and evaluate the effectiveness of different dialysis treatments. In the past years, a large number of strategies aimed at lowering the level of serum IS, such as diet modification and the administration of charcoal and biotic supplements, have been suggested. Hence, the detection of this toxin may offer a screening method for the CKD population at high risk of CV events. Moreover, the short-time detection of gut-derived indoxyl sulfate would help clinicians assess the effects of several treatments aimed at lowering uremic toxins and predict the outcome of renal dysfunction. 

## Figures and Tables

**Figure 1 ijms-24-05142-f001:**
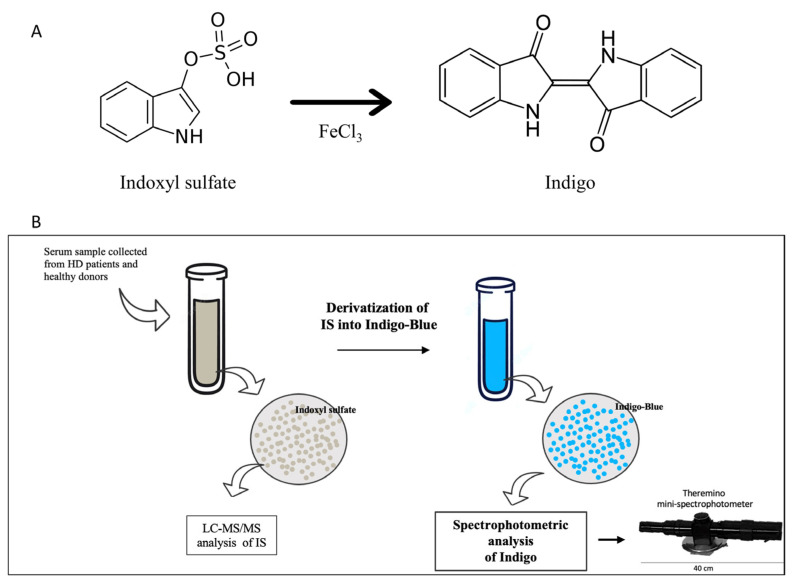
(**A**) Process of IS conversion into Indigo. IS-derived Indigo is a chromophore that could be detected in UV-VIS mode by spectrophotometric analysis (Theremino) and LC-MS/MS. (**B**) Experimental design. Firstly, the total concentration of IS in serum was investigated by tandem mass spectrometry. Next, the whole fraction of IS in each sample was converted into Indigo by derivatization. Finally, the concentration of Indigo was detected by spectrophotometry using the mini-spectrophotometer Theremino.

**Figure 2 ijms-24-05142-f002:**
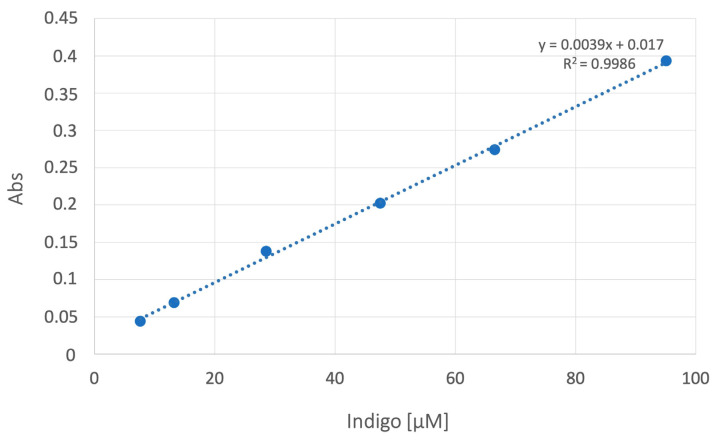
Calibration curve. The linear correlation between absorbance and Indigo concentrations was between 7.5 and 95 µM. Regression analysis showed R^2^ ≥ 0.99.

**Figure 3 ijms-24-05142-f003:**
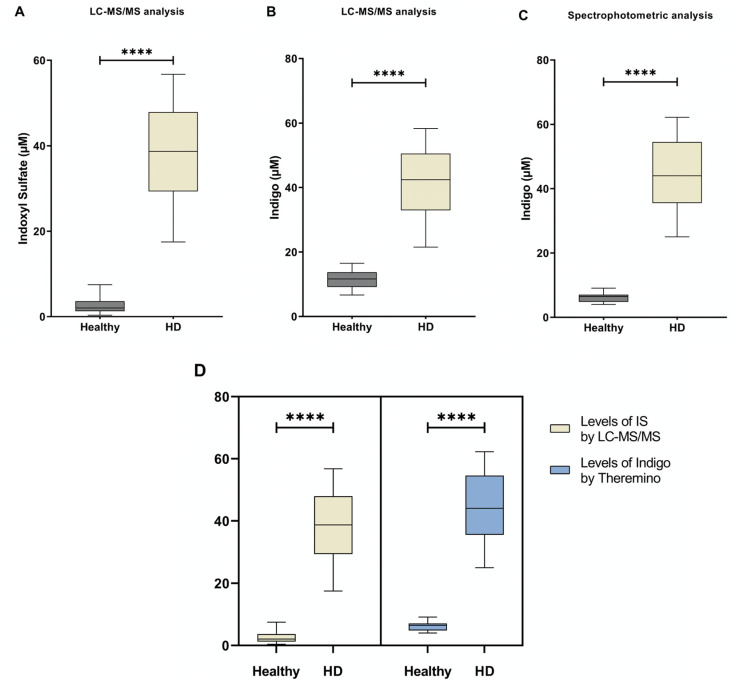
The level of IS and Indigo blue measured by LC-MS/MS and Theremino. (**A**) Differences between IS concentrations obtained in the two groups by using the standard LC-MS/MS approach. (**B**) Differences in Indigo levels between groups obtained by LC-MS/MS. (**C**) Differences in Indigo levels obtained by Theremino. (**D**) No differences were found in IS and Indigo levels among groups obtained by LC-MS/MS and a Theremino spectrophotometer, respectively. **** *p* < 0.001. Abbreviation: HD, Hemodialysis; IS, indoxyl sulfate.

**Figure 4 ijms-24-05142-f004:**
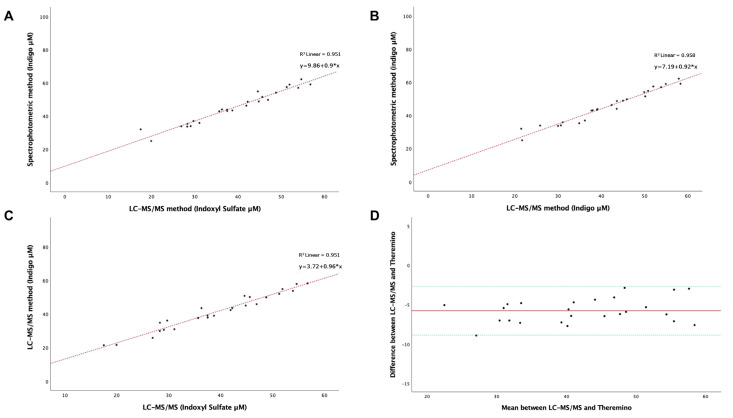
(**A**–**C**) Linear regression analysis of the relationship between the LC-MS/MS method and colorimetric assay performed using Theremino on IS and IS-derived Indigo in HD patients. (**D**) Evaluation of the agreement by the Bland–Altman plot using the difference and the mean among the data of the two methods. Upper and lower green lines represent the region of agreement, and the red line represents the bias between methods (−5.7732) obtained by the mean of the difference of results.

**Figure 5 ijms-24-05142-f005:**
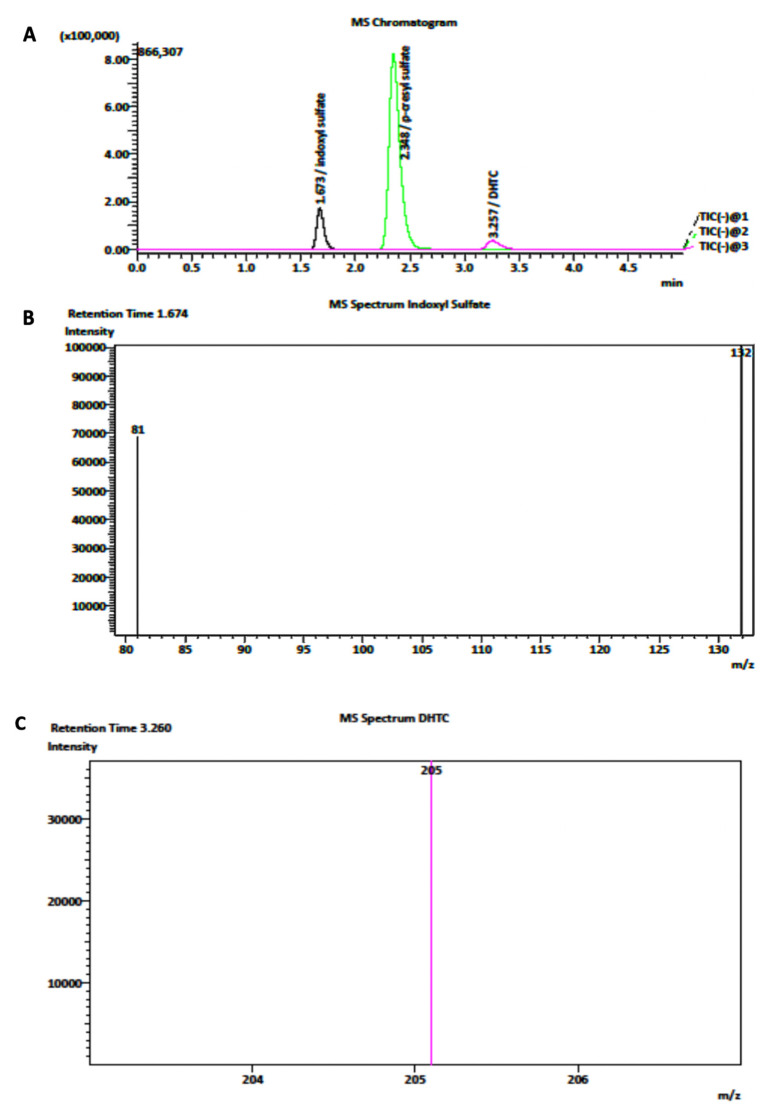
(**A**) Representative chromatograms of IS, DHTC, and pCS in the MRM mode with negative-mode ionization. (**B**,**C**) Monitored fragmentation of respectively IS and DHTC, respectively.

**Table 1 ijms-24-05142-t001:** Liquid chromatography–tandem mass spectrometry settings.

Analyte	Molecular Weight (g/mol)	Retention Time (min)	Ionization Mode (ESI)	Precursor Ion (m/z)	Product Ion (m/z)	Collision Energy (eV)
IS	213.21	1.51	Negative	212.10	151.9	18
pCS	187.05	2.25	Negative	187.05	107.0	23
DHTC	297.741	3.25	Negative	296.15	205.10	24
Indigo	262.27	3.8	Negative	261.0	157.05	32

Analytical parameters used for the analysis and collision energies of the analyte. Abbreviation: ESI (electrospray ionization), IS (indoxyl sulfate), pCS (para-cresyl sulfate), DHTC (hydrochlorothiazide), and Indigo.

## Data Availability

Not applicable.

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
