# Peer review of "Gut-Derived Uremic Toxins in CKD: An Improved Approach for the Evaluation of Serum Indoxyl Sulfate in Clinical Practice"

_ijms, 2023, doi:10.3390/ijms24065142_

Round 1
Reviewer 1 Report
The research showed novel method of IS measurement with Theremino spectrophotometer, which provided the real-time assessment of IS levels with cost reduction. I am afraid of whether this method sustains high reproducibility, for example with cloudy serum.
Author Response
We thank the Reviewer for pointing this out and agree that we should clarify this aspect in the manuscript. Therefore we added the following statement in the "Stability" section of Methods:
"IS quantification with Theremino is based on spectrophotometry, thus relying on the capacity of IS to originate Indigo. Based on our analyses we observed that the sensitivity of the Theremino was not dependent on blood rheology (e.g. cloudy serum or blood clots) as there is no interference with IS concentration or derivatization reaction. Moreover, we observed that sample quality did not influence IS reactions such as Indigo generation or its absorbance at wavelength 420-450 nm and evaluated the ability of Theremino to perform the quantification under different conditions such as freeze/thaw cycles."
Reviewer 2 Report
The authors presented a colorimetric assay to quantify the uremic toxin indoxyl sulfate in serum. The method has several advantages when compared with LC-MS (low cost, easy interpretation) that can be important in a clinical setting.
However, in my opinion, the paper presents some important limitations.
1. The authors explain very little about the origin and roles of indoxyl sulfate. This information has to be presented in the introduction.
2. The methodology presents some issues that I think should be addressed:
Quantification of free and protein-binding fractions. Authors explain in the introduction that IS accumulates mainly due to the impossibility of the kidneys to filtrate protein-bound IS. Thus, the methodology should be able to quantify both the free and protein-bound fractions. Authors are encouraged to quantify i) the total fraction through derivatization before protein precipitation; ii) free fraction through derivatization after protein precipitation; iii) protein-bound fraction by subtraction (total-free). Other methods using proteinase K have been also described to cleave the binding IS-albumin (DOI:10.1038/s41598-018-27983-0) to quantify the total fraction.
I am also concerned about the specificity of the method. Other tryptophan metabolites that contain an indol moiety might also derivatize. This needs to be evaluated.
Inter-day and intra-day variability were not assessed.
Moreover, the authors do not explain how calibration curves were performed. Which solvent did the authors used to prepare the concentrations? Did the authors evaluate it here was a matrix effect?
There is no reference of the mini-spectrophotometer Theremino that was used.
Finally, the sample size is not enough to publish a method.
3. There is no information and/or ethics statement regarding the collection of the clinical samples.
Author Response
Dear Reviewer,
We are grateful for such constructive remarks and useful suggestions, which have significantly improved the manuscript's quality. We have carefully evaluated the comments and addressed them changing the manuscript accordingly.
- The authors explain very little about the origin and roles of indoxyl sulfate. This information has to be presented in the introduction.
REPLY: Thank you for this observation. We have elaborated this content in the introduction of the manuscript (page 1-2, line 36 – 64).
- The methodology presents some issues that I think should be addressed: Quantification of free and protein-binding fractions. Authors explain in the introduction that IS accumulates mainly due to the impossibility of the kidneys to filtrate protein-bound IS. Thus, the methodology should be able to quantify both the free and protein-bound fractions. Authors are encouraged to quantify i) the total fraction through derivatization before protein precipitation; ii) free fraction through derivatization after protein precipitation; iii) protein-bound fraction by subtraction (total-free). Other methods using proteinase K have been also described to cleave the binding IS-albumin (DOI:10.1038/s41598-018-27983-0) to quantify the total fraction.
REPLY: Thank you for this observation. Our study aimed at developing a smart kit which allows to rapidly quantify the total IS in a serum sample for clinical practice. One of the main advantages of our method is the rapid test execution thanks to the possibility to skip several steps, such as the detection of bound-fraction. The detection of both free IS and protein-bound IS in clinical practice would require at least two serum sample from the same patient. Moreover, the quantification with the spectrophotometer method aims to help clinicians address the efficiency of blood purification from uremic toxins (e.g., dialysis, nutritional treatments of pharmacologic therapies) in the context of chronic kidney diseases and thus in relative terms of clearance.
I am also concerned about the specificity of the method. Other tryptophan metabolites that contain an indol moiety might also derivatize. This needs to be evaluated. Inter-day and intra-day variability were not assessed.
Moreover, the authors do not explain how calibration curves were performed. Which solvent did the authors used to prepare the concentrations? Did the authors evaluate it here was a matrix effect?
REPLY: Our reaction is specific for the indoxyl sulfate since the derivatization reaction of IS to indigo involves the exact stoichiometry and the structure of the molecule. This means that normally, metabolites that contain indole group can be derivatized; however, a unique way exist to derivatize IS in indigo, and this involve the reagents (IS and FeCl3) and molecule stoichiometry. Under these conditions, only the indoxyl sulfate is able to react with FeCL3 to generate a molecule (indigo) with absorbance property.
(i) We have revised the text to address these concerns and better clarify these aspects. According to your suggestions, all measurements for defining the linearity and the accuracy have been reported in the additional paragraph 2.3. (page 4, line 128-147).
(ii) The solvent we used to prepare both the standards and samples was the acetonitrile extraction solvent. We added this information to the manuscript.
(iii) Our method is based on the analyte extraction and derivatization in acetonitrile. Therefore, the matrix effect is nullified and does not interfered with the derivatization of indoxyl sulfate as acetonitrile is inert and is undetected by both the LC-MS/MS and spectrophotometer.
There is no reference of the mini-spectrophotometer Theremino that was used.
REPLY: References for Theremino are not possible as it is the first prototype. To date, a smart spectrophotometer able to read the indigo at 420 nm is not available.
Finally, the sample size is not enough to publish a method.
REPLY: This is a pilot study because the spectrophotometer is still a prototype. So our goal was to describe the property of the method on the basis of the results obtained in our analysis. Essentially the sample size is a limitation of our work, however it could represent a first step for its implementation. We stated this limitation in the discussion.
- There is no information and/or ethics statement regarding the collection of the clinical samples.
REPLY According to your suggestions, we have added information concerning the enrollment criteria, ethics approval number, and patient’s data to the 5.1 section of the manuscript (page 8, line 270 – 278)
We would like to sincerely express our gratitude for the thorough revision of our manuscript. Thank you for your time.